# Odour Emissions of Municipal Waste Biogas Plants—Impact of Technological Factors, Air Temperature and Humidity

**Marta Wiśniewska \*** , **Andrzej Kulig and Krystyna Lelicińska-Serafin**

Hydro and Environmental Engineering, Faculty of Building Services, Warsaw University of Technology, 20 Nowowiejska Street, 00-653 Warsaw, Poland; Andrzej.Kulig@is.pw.edu.pl (A.K.); Krystyna.Lelicinska@pw.edu.pl (K.L.-S.)
**\*** Correspondence: Marta.Wisniewska.89@wp.pl

**Abstract:** Biogas plants processing municipal waste are an important part of a circular economy (energy generation from biogas and organic fertiliser production for the treatment of selectively collected biowaste). However, the technological processes taking place may be associated with odour nuisance. The paper presents the results of pilot research conducted at six municipal waste biogas plants in Poland. It shows the relations between odour intensity and concentration and the occurring meteorological and ambient conditions (air temperature and relative humidity) and technological factors at biogas plants processing municipal waste. The impact of meteorological and ambient conditions was identified by measuring air temperature and relative humidity and observing their changes. The impact of technological factors was identified by measuring odorant concentration (volatile organic compounds and ammonia) and observing their changes between individual measurement series. At most analysed biogas plants, the influence of technological factors on odour emissions took place and was clearly noted. The elements of biogas installations characterised by the highest concentration of these odorants were indicated. Special attention should be paid to the choice of technological solutions and technical and organisational measures to reduce the impact of unfavourable atmospheric conditions on odour emissions.

**Keywords:** ambient conditions; ammonia; biogas plant; biowaste; environmental conditions; meteorological conditions; municipal waste treatment; odour concentration; odour intensity; odour source

## 1. Introduction

Waste treatment plants, which are part of municipal infrastructure, are very significant from the point of view of urban development. Waste management objects, including mechanical–biological treatment plants (MBT), are an indispensable part of the municipal infrastructure [1,2]. Municipal solid waste has the potential for recovery of materials and energy and can be a renewable source [3–6]. Due to the improvement in the amount of newly built mechanical–biological treatment plants in Europe, the qualitative characteristics of odour emissions from these investments is becoming a requisite for the attention of appropriate planning of future installations, and also protection of health of both their operating employees and people living in areas close to the plants [7,8]. These reasons contributed to the publication of scientific papers presenting measurements and chemical characteristics of emissions from waste treatment plants [8–10]. Some of the above-mentioned works focus on the investigation of parameters that bring to the development of unpleasant smells into focus [8,9].

Odour emissions pose trouble due to their influence over public health through the nuisance of people [11]. The odorants, which are the chemical compounds causing a characteristic unpleasant smell,

are naturally associated with potential danger, arduousness and can result in damaging psychosomatic effects. Therefore, the emission of gases polluted by malicious substances into the atmosphere is a meaningful environmental problem [12–14]. So far, the research into the field of odours has mainly focused on sewage treatment plants [15], swim housing [16], landfills [17] and farms with animal husbandry and biogas facilities [18]. Orzi et al. [11] identified the need to find relationships between biological waste treatment processes and environmental impacts, including odour emissions; but so far, research on odour emission factors for fermentation processes and composting of municipal waste is rare [19,20].

Municipal waste biogas plants make up a part of MBT installations. Either the mechanical or biological parts constitute sources of odour emissions [21]. Chemical compounds with a nuisance can be emitted during the various unit operations, storage and transport of collected waste, mechanical processing for preparation to the next processes, biological processing in anaerobic conditions, dewatering of digestate and also aerobic stabilisation [2]. According to Lapčík and Lapčíková [22], who analysed biogas plants located on the territory of Czech Republic, one of the primary kinds of impact of investments on the environment is the emission of various pollutants; for instance, odours. These researchers came to the conclusion that one of the most important causes of odour emission is the incorrect organisation and also lack of encapsulation of technological processes. Odour emission in the waste treatment depends on various factors, for instance: the type of the raw material (waste subject to the processes), processing methods, applied design solutions and the operating method of the installation, including the technological regime. A previous paper [23] presents the results of the analysis of the impact of technologies used at two biogas plants located in Poland, as well as the observance of the technological regime. Municipal mixed waste produced in households is characterised by a high content of biodegradable fractions. Process gases, which are released during the above operations contain organic (volatile organic compounds, which include ketones, alkanes, alkenes, alcohols, acids, terpenes and also organic sulphur compounds) and inorganic compounds ($H_2S$, $NH_3$). These chemical compounds may constitute a significant source of odour nuisance either in cities or in villages and pose a threat to public health and safety of the natural environment [24–27]. Ammonia is particularly dangerous for the eyes and respiratory system [28,29]. This compound has the ability to form both organic and inorganic aerosols, which can increase the amount of dust in the air [29].

Meteorological conditions for waste collection, storage and processing also have a significant impact from the point of view of odour emission [21]. The literature review has shown that the relationship between these factors is not yet well understood. The results of researches published so far do not give a clear answer on the relationship between meteorological and ambient conditions, odorant concentrations, odour concentrations and waste treatment, because most often they were carried out in a short period of time, or the scientists did not examine all the parameters at the same time. Knowledge of the effect of the above factors will permit the application of efficient operation measures, minimising the odour nuisance of the installations for areas close to the plants.

The most important elements influencing the changes taking place in the pollution cloud over the areas of waste management facilities are: the degree of vertical air stability, wind direction and speed, the character of air turbulence and precipitation. In addition, the temperature and humidity of the air, the presence and possibility of precipitation, as well as the pressure and degree of cloudiness must also be taken into account [30,31].

An increase in wind speed causes a decrease in the concentration of components in a cloud of pollutants. Therefore, the wind speed parameter has a positive influence on the decrease of pollutant concentration. The highest concentrations of pollutants are observed in the lowest ground level air layer. As the distance from the emission source increases, concentrations in the upper atmosphere layers increase and decrease in the lower layers [32]. The contaminants have the ability to shift in the atmospheric layer together with the surrounding air masses, adopting a speed equal to the wind speed.

Precipitation and humidity of atmospheric air also affect the range of the cloud of pollutants. The occurrence of rain reduces the concentration of pollutants due to the phenomenon of absorption on the surface of drops.

High summer temperatures result in higher odour emissions from surface sources. Fragrance emissions increase during the summer not only due to increased variability, but also due to the increase in anaerobic activity of the bacteria. In addition, oxygen is less soluble at higher temperatures, so the conditions usually become anaerobic. Although high summer temperatures cause higher odorant emissions from open sources, unstable weather conditions during this period tend to disperse pollutants easily. In winter, the situation is reversed. At lower temperatures, emissions are lower, but often also less dispersed due to stable weather conditions [33,34].

Szulczyński et al. [35] carried out research in Poland around three various sources of odour nuisance (one of the analysed sources was related to waste management) in the six-month period from January to June. They conclude that the biggest odour concentrations around each of the analysed sources were observed in the summer months. In terms of their location, the biggest odour concentrations occurred in the vicinity of the municipal landfill. This research shows that values of mean odour concentration decrease with an increase in relative air humidity in the surroundings, which may result from the phenomenon of sorption of odour compounds on the surface and within the droplets of water mist, bringing about a decrease in the odour concentration in the surrounding air. Wilson and Baieto [36] conducted similar research and obtained similar results.

This paper presents the results of a pilot study conducted at six biogas plants processing municipal waste. The aims of this study were: to record odour concentration ($c_{od}$) and intensity ($i_{od}$); to analyse odour emissions in relation to selective atmospheric and ambient conditions (air temperature and relative humidity inside and outside of technological facilities); to record ammonia and volatile organic compounds (VOC) concentrations; and to find relations between odour concentration and ammonia and VOC concentrations. The influence of technological factors was also taken into account by conducting measurements in various sources of odour emissions associated directly with the technological process. Research was performed at various odour sources differentiated according to the technological solutions used at the analysed biogas plants.

This knowledge is very significant from the point of view of the operation and exploitation of both existing installations and those that probably arise in the future. Knowledge of the effect of meteorological and ambient conditions and odorant concentrations related to the waste treatment on the odour impact of waste treatment plants will permit the application of efficient operation measures minimising the odour nuisance of the installations for areas close to the plants.

The results of researches published so far do not give a clear answer on the relationship between the above-mentioned factors, because most often they were carried out in a short period of time, or the scientists did not examine all the parameters at the same time. This paper makes up a substantial contribution to the field and make up a part of more detailed research aimed at the determination of dependencies between odour and odorant concentration, odour intensity and also a few different factors, either meteorological and ambient or technological.

## 2. Materials and Methods

### 2.1. Study Methodology

The study covers two measurement series (1 and 2) at six plants (A, B, C, D, E, F). It was carried out at the turn of 2018 and 2019. The dates of the series are presented in Table 1. The measurements always took place in various seasons, always during the day. The research involves the determination of air temperature and relative humidity at each designated measurement point (receptors marked as a, b, c, d, e, f, g), simultaneously measuring wind speed and direction at points located outside buildings and plant processing buildings. Meteorological conditions were determined in the plume at a height of 1.5 m by means of a Weather Meter 4500 NV by Kestrel based on the European Standard VDI 3940 [37].

The results of air temperature and relative humidity presented in Table 5 are the arithmetic mean of five parallel measurements. Furthermore, in each receptor, odour intensity ($i_{od}$) was determined by means of the method of sensory evaluation in a six-grade scale, in accordance with Table 2 [31], as well as the odour concentration ($c_{od}$) expressed in odour units per cubic metre ($ou/m^3$), analogically to European norm PN-EN 13725:2007 [38]. Olfactometric tests were performed in situ with the use of dynamic field olfactometric measurements. The device used in the study was the olfactometer Nasal Ranger® (St. Croix Sensory, Inc., Stillwater MN, USA]). Odour concentrations in every measurement point were calculated based on two readings of the *D/T* parameter (dilution-to-threshold ratio), using the following formulas [2,31,35,39,40]:

$$Z_{YES} = (D/T)_{YES} + 1,$$

where $Z_{YES}$ means dilution ratio, at which the odour was perceptible (-); $(D/T)_{YES}$ means dilution ratio agreeing for the moment when the odour was perceptible for the first time (-),

$$Z_{NO} = (D/T)_{NO} + 1,$$

where $Z_{NO}$ means dilution ratio, at which the odour was imperceptible; $(D/T)_{NO}$ means dilution ratio agreeing for the moment when the odour was imperceptible just before the dilution $(D/T)_{YES}$,

$$Z_{ITE} = \sqrt{Z_{YES} \cdot Z_{NO}},$$

where $Z_{ITE}$ means the assessment of the individual threshold, expressed as dilution ratio (-). Values of the odour concentrations were calculated based on a geometric mean of the set of all individual estimations ($Z_{ITE}$) for a given measurement point:

$$c_{od} = \sqrt[n]{\sum_{i=1}^{n} Z_{ITE,i}},$$

where *n* means amount of all estimates.

**Table 1.** Schedule of measurement series at the analysed biogas plants.

| Location of Biogas Plant | Date of 1st Series | Date of 2nd Series |
|---|---|---|
| Jarocin | 21 July 2018 | 27 February 2019 |
| Tychy | 27 September 2018 | 5 March 2019 |
| Promnik | 12 April 2019 | 15 May 2019 |
| Stalowa Wola | 30 August 2018 | 19 February 2019 |
| Wólka Rokicka | 28 August 2018 | 19 February 2019 |
| Biała Podlaska | 26 April 2019 | 15 May 2019 |

**Table 2.** Scale of odour intensity according to the sensory evaluation method [31].

| Scale of Odour Intensity, $i_{od}$ | Discernible Odour Intensity |
|---|---|
| 0 | No odour |
| 1 | Odour almost imperceptible |
| 2 | Very weak odour |
| 3 | Weak odour |
| 4 | Strong odour |
| 5 | Very strong odour |

Chemical tests, including the determination of concentrations of volatile organic compounds (VOC) and ammonia ($NH_3$), were performed with the MultiRae Pro gas detector (RAE Systems, Inc., San Jose, CA, USA]). The sensor's characteristic is presented in Table 3. At each of the measurement

points, measurements of the analysed compounds were made in five repetitions. The ionisation energy of the sensor was 10.6 eV. The sensor detects only those compounds that are able to ionise [41,42]. The measuring error of the device is 10 ppb for VOC and 1 ppm for ammonia.

**Table 3.** Characteristic of the gas detector sensors.

| Kind of Sensor | Type of Sensor | Resolution | Range |
|---|---|---|---|
| NH$_3$ | Electrochemical | 1 ppm | 0–100 |
| VOC | Photoionisation (PID) | 0.01 ppm | 0–100,000 ppm |

*2.2. Characteristics of the Analysed Plants*

The biogas plants included in this study are located in Poland and cover the western, southern and eastern regions of the country in the following municipalities: A, Jarocin; B, Tychy; C, Promnik; D, Stalowa Wola; E, Wólka Rokicka; and F, Biała Podlaska. Their locations are presented in Figure 1.

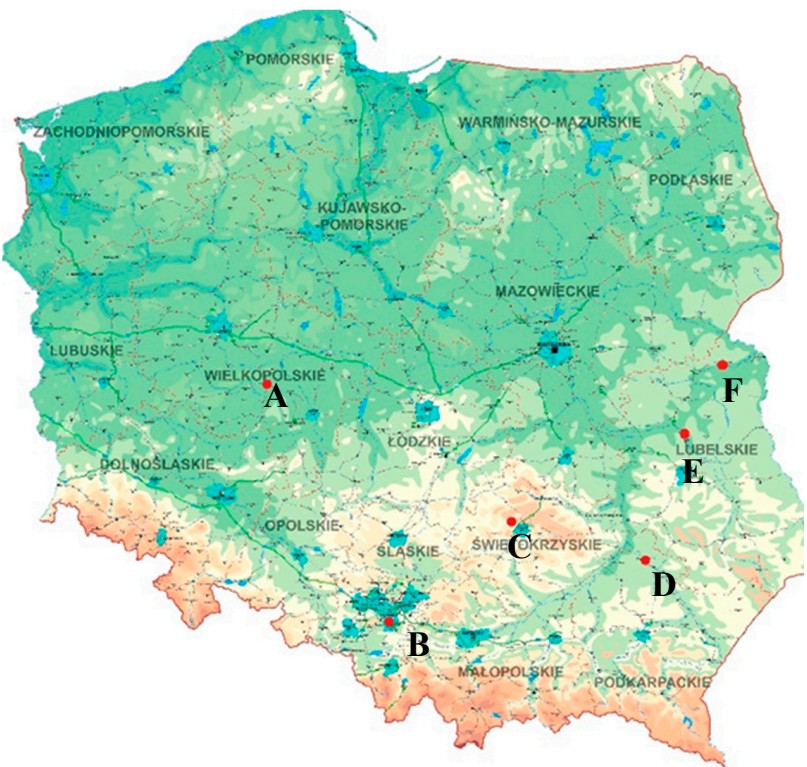

**Figure 1.** Location of biogas plants processing municipal waste in Poland (letters from A to F designate the installations included in the study) (based on [43]).

The analysed biogas plants consist of a mechanical part (waste pre-treatment) and a biological part (methane fermentation). In addition, digestate and biogas are processed at the plants. At most analysed installations, mixed municipal waste is treated. In one of them (E biogas plant in Biała Podlaska), biowaste from a selective collection is processed. The mechanical part is similar at every biogas plant. Then, biodegradable fractions are treated biologically under anaerobic conditions using dry or semi-dry fermentation. The dry fermentation process takes place exclusively at E biogas plant (in Wólka Rokicka). In this technology, unlike other biogas plants, the fermentation process takes place without digestate dewatering.

### 2.3. Odour Sources in Municipal Waste Biogas Plants

At the examined biogas plants, seven main sources (marked with letters a–g) of odour emissions are identified. They are presented in Table 4.

**Table 4.** Odour sources identified at the examined biogas plants.

| Mark of Odour Source | Name of Odour Source at Analysed Biogas Plants | | | | | |
|---|---|---|---|---|---|---|
| | A | B | C | D | E | F |
| a | Waste storage plant | | | | | |
| b | Mechanical part plant (pre-treatment) | | | | | |
| c | Fermentation preparation plant/field | | | | | |
| d | Digestate dewatering plant | | | | — | — |
| e | — | — | Oxygen stabilisation plant | | — | Oxygen stabilisation plant |
| f | Oxygen stabilisation | — | Oxygen stabilisation | | | |
| g | Open biofilter surface | | | | | |

Notes: "—" sources do not occur on the premises of a given plant, due to the applied waste treatment technology or practiced technological regime.

## 3. Results and Discussion

### 3.1. Atmospheric and Ambient Conditions

The results of the relationship between atmospheric and environmental conditions in the identified sources of odours in the individual measurement series are shown in Table 5.

**Table 5.** Measurements of air temperature (T) and relative humidity (RH) obtained in the pilot research at six biogas plants processing municipal waste; values at individual odour sources and mean value with standard deviation.

| Biogas Plant | Parameter | No. of the Research Series | Odour Sources Identified at the Examined Biogas Plants | | | | | | |
|---|---|---|---|---|---|---|---|---|---|
| | | | a | b | c | d | e | f | g |
| A | T [°C] | 1 | 31.0 | 32.0 | 28.7 | 30.5 | — | 27.7 | 32.0 |
| | | | Mean 30.3 ± 1.8 | | | | | | |
| | | 2 | 10.9 | 11.7 | 11.4 | 11.8 | — | 13.0 | 12.6 |
| | | | Mean 11.9 ± 0.8 | | | | | | |
| | RH [%] | 1 | 53.0 | 51.0 | 52.5 | 57.5 | — | 34.3 | 31.4 |
| | | | Mean 46.6 ± 10.9 | | | | | | |
| | | 2 | 73.4 | 73.4 | 73.4 | 73.4 | — | 65.1 | 56.3 |
| | | | Mean 69.2 ± 7.1 | | | | | | |
| B | T [°C] | 1 | 16.9 | 16.9 | 16.0 | 16.5 | — | — | 16.0 |
| | | | Mean 16.5 ± 0.5 | | | | | | |
| | | 2 | 18.9 | 13.5 | 12.4 | 14.2 | — | — | 10.2 |
| | | | Mean 13.8 ± 3.2 | | | | | | |
| | RH [%] | 1 | 48.0 | 42.5 | 48.5 | 49.0 | — | — | 55.5 |
| | | | Mean 48.7 ± 4.6 | | | | | | |
| | | 2 | 42.6 | 39.5 | 54.4 | 63.5 | — | — | 49.1 |
| | | | Mean 49.8 ± 9.6 | | | | | | |

**Table 5.** *Cont.*

| Biogas Plant | Parameter | No. of the Research Series | Odour Sources Identified at the Examined Biogas Plants | | | | | | |
|---|---|---|---|---|---|---|---|---|---|
| | | | a | b | c | d | e | f | g |
| C | T [°C] | 1 | 16.0 | 16.8 | 17.6 | 16.5 | nm | 9.6 | nm |
| | | | Mean 15.3 ± 2.9 | | | | | | |
| | | 2 | 17.5 | 17.3 | 16.2 | 17.8 | nm | 9.3 | nm |
| | | | Mean 15.6 ± 3.2 | | | | | | |
| | RH [%] | 1 | 55.5 | 47.4 | 49.4 | 58.0 | nm | 43.6 | nm |
| | | | Mean 50.9 ± 5.3 | | | | | | |
| | | 2 | 66.0 | 57.6 | 65.4 | 62.0 | nm | 72.8 | nm |
| | | | Mean 64.8 ± 5.0 | | | | | | |
| D | T [°C] | 1 | 26.5 | 26.3 | 25.5 | 25.3 | 25.2 | 27.1 | 27.0 |
| | | | Mean 26.1 ± 0.8 | | | | | | |
| | | 2 | 16.3 | 16.7 | 16.0 | 15.7 | 15.0 | 12.3 | 13.0 |
| | | | Mean 15.0 ± 1.7 | | | | | | |
| | RH [%] | 1 | 50.8 | 50.5 | 59.3 | 59.3 | 53.5 | 43.5 | 41.5 |
| | | | Mean 51.2 ± 7.0 | | | | | | |
| | | 2 | 36.8 | 36.3 | 45.3 | 41.9 | 42.8 | 46.5 | 73.2 |
| | | | Mean 46.1 ± 12.6 | | | | | | |
| E | T [°C] | 1 | 25.6 | 25.0 | 26.0 | — | — | 21.0 | 22.4 |
| | | | Mean 24.0 ± 2.2 | | | | | | |
| | | 2 | 13.5 | 15.8 | 11.6 | — | — | 15.2 | 12.3 |
| | | | Mean 13.7 ± 1.8 | | | | | | |
| | RH [%] | 1 | 51.8 | 62.9 | 63.2 | — | — | 61.0 | 55.6 |
| | | | Mean 58.9 ± 5.0 | | | | | | |
| | | 2 | 43.6 | 42.6 | 55.6 | — | — | 43.5 | 47.3 |
| | | | Mean 46.5 ± 5.4 | | | | | | |
| F | T [°C] | 1 | 25.2 | 25.2 | 28.7 | — | 23.5 | 26.4 | nm |
| | | | Mean 25.8 ± 3.0 | | | | | | |
| | | 2 | 14.8 | 14.8 | 11.7 | — | 14.2 | 11.7 | nm |
| | | | Mean 13.4 ± 2.1 | | | | | | |
| | RH [%] | 1 | 40.0 | 40.9 | 33.2 | — | 58.4 | 33.3 | nm |
| | | | Mean 41.2 ± 9.2 | | | | | | |
| | | 2 | 86.6 | 83.6 | 100 | — | 96.5 | 100 | nm |
| | | | Mean 93.3 ± 6.9 | | | | | | |

Notes: "—" sources do not occur at the premises of a given plant, due to the applied waste treatment technology, "nm" not measured.

During the research at four biogas plants, two measurement series were performed with significantly different average air temperatures. Temperature variations range from 10.3 °C in Wólka Rokicka to 18.4 °C in Jarocin together with 11.1 °C in Stalowa Wola and 12.4 °C in Biała Podlaska. At two other biogas facilities (in Tychy and Promnik), differences between average air temperatures during the implementation of two measurement series are small at: 3.0 °C and 0.3 °C, respectively. The mean relative humidity of air was variable in the case of research carried out at four biogas plants:

at the level of 52.1% in Biała Podlaska, 22.6% in Jarocin, 13.9% in Promnik and 12.4% in Wólka Rokicka. At other biogas plants (Tychy and Stalowa Wola), the air humidity was very similar during the period of the research (approximately 50%). Differences in air temperature and humidity between series are caused by measurements being carried out at different times (Polish climate is characterised by transience through the influence of polar-sea and polar-continental air masses). The biggest differences are noted especially in Jarocin, Stalowa Wola and Wólka Rokicka, where the measurement series were led at other time of the year (Table 1).

### 3.2. Odour Concentration ($c_{od}$) and Intensity ($i_{od}$)

Figures 2–7 show the results of odour concentration ($c_{od}$) and intensity ($i_{od}$) at individual odour sources (a–g) at the analysed biogas plants (for two measurement series: 1 and 2) in combination with the average temperature (T) and average relative humidity of air (RH). Air temperature and humidity values indicated in the figures are the arithmetic mean of the values measured at individual measuring points during one measurement series.

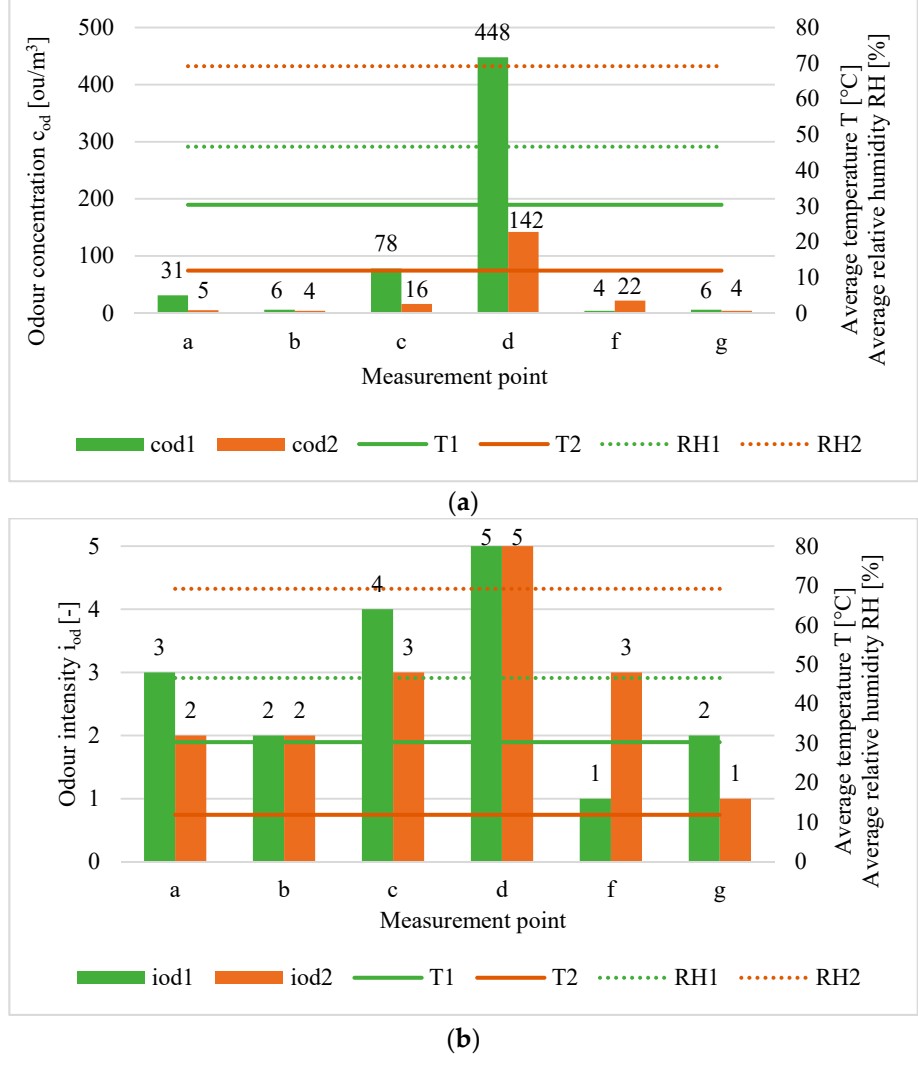

**Figure 2.** Dependencies between average air temperature and relative humidity and $c_{od}$ and $i_{od}$ in two measurement series at Jarocin biogas plant (21 July 2018 and 27 February 2019), (**a**) T, RH and $c_{od}$; (**b**) T, RH and $i_{od}$.

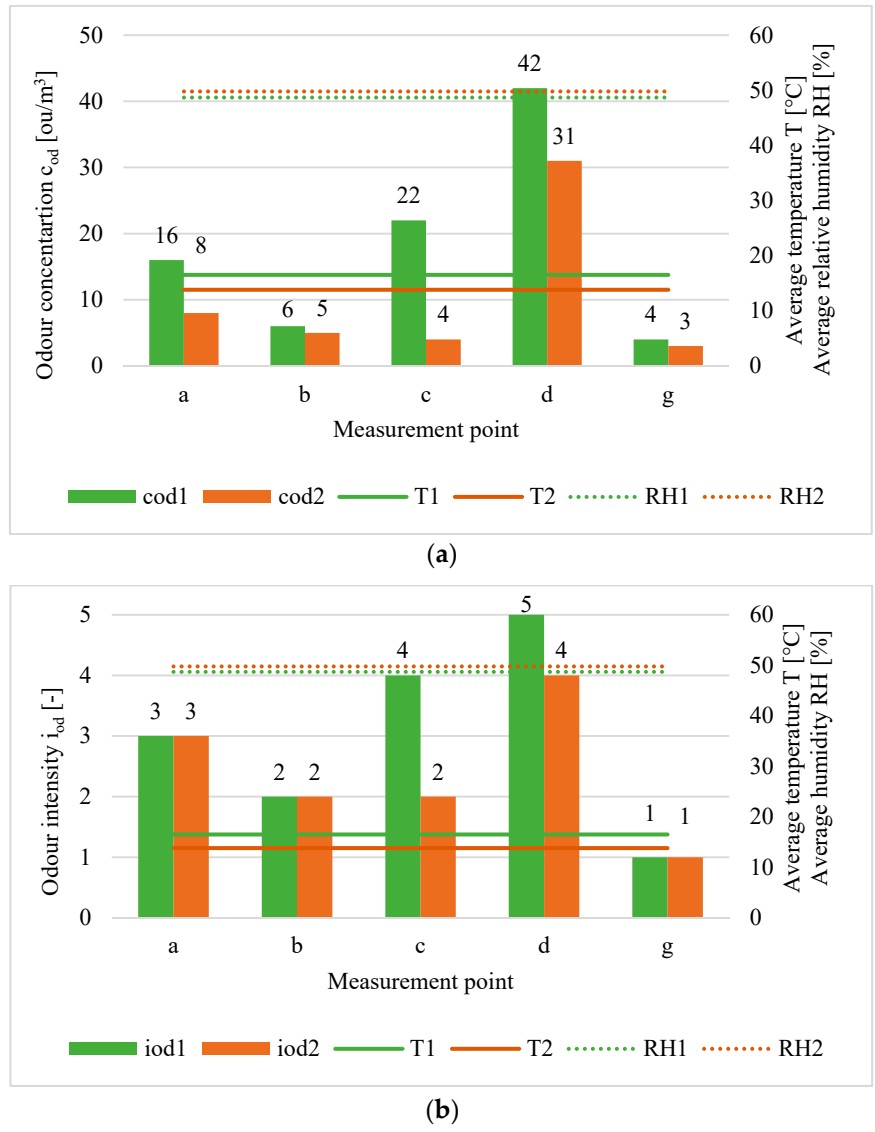

**Figure 3.** Dependencies between average air temperature and relative humidity and $c_{od}$ and $i_{od}$ in two measurement series at Tychy biogas plant (27 September 2018 and 5 March 2019), (**a**) T, RH and $c_{od}$; (**b**) T, RH and $i_{od}$.

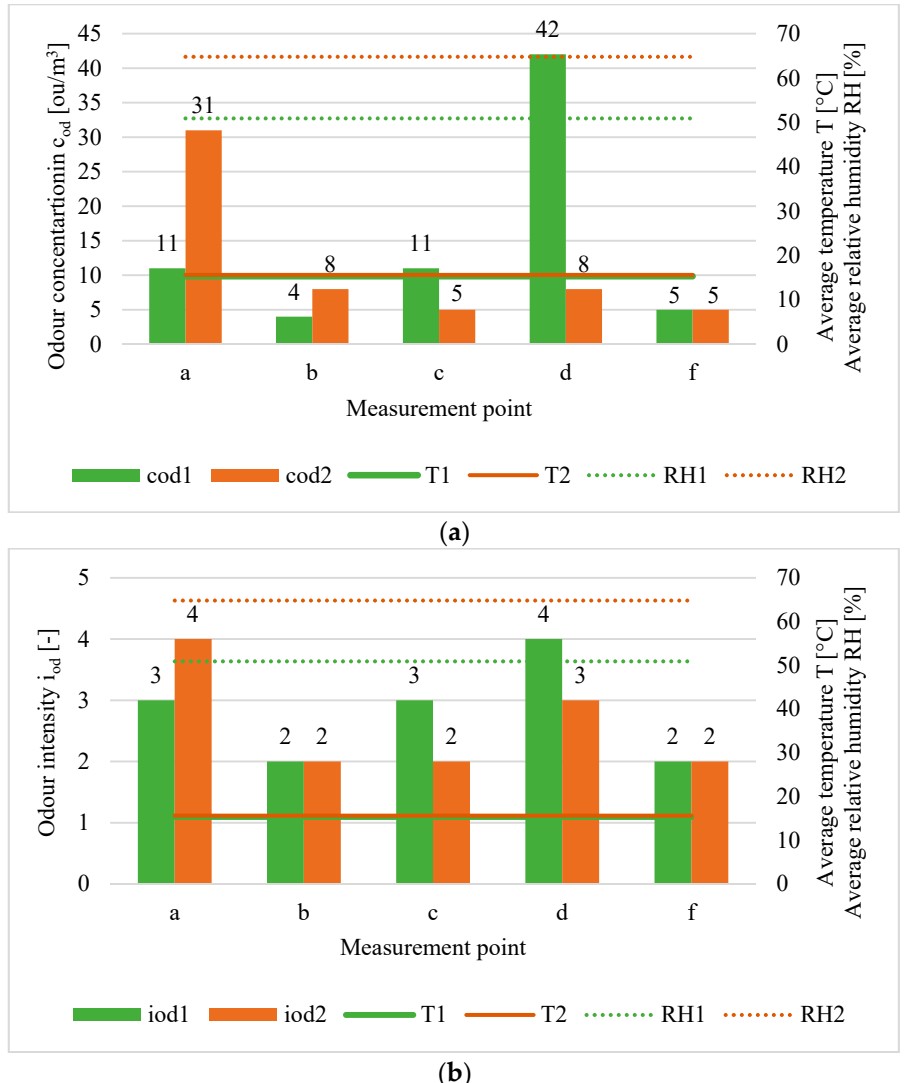

**Figure 4.** Dependencies between average air temperature and relative humidity and $c_{od}$ and $i_{od}$ in two measurement series at Promnik biogas plant (12 April 2019 and 14 May 2019), (**a**) T, RH and $c_{od}$; (**b**) T, RH and $i_{od}$.

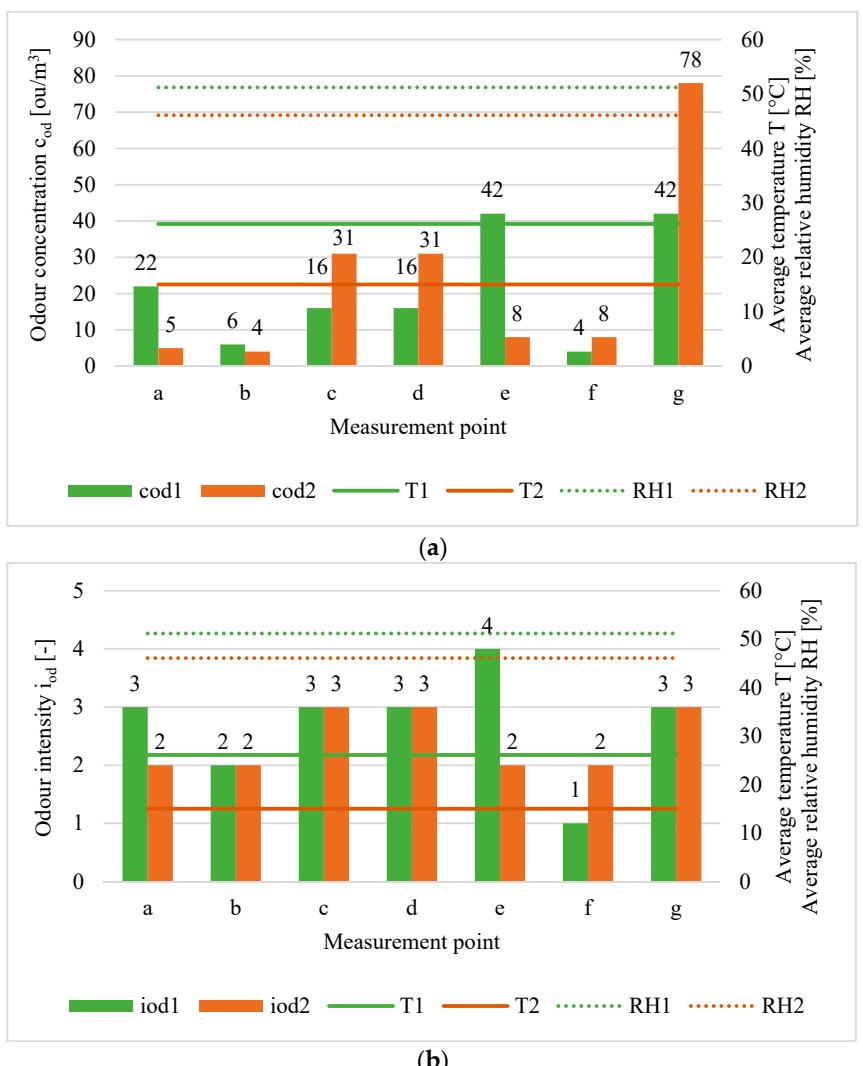

**Figure 5.** Dependencies between average air temperature and relative humidity and $c_{od}$ and $i_{od}$ in two measurement series at Stalowa Wola biogas plant (30 August 2018 and 20 February 2019), (**a**) T, RH and $c_{od}$; (**b**) T, RH and $i_{od}$.

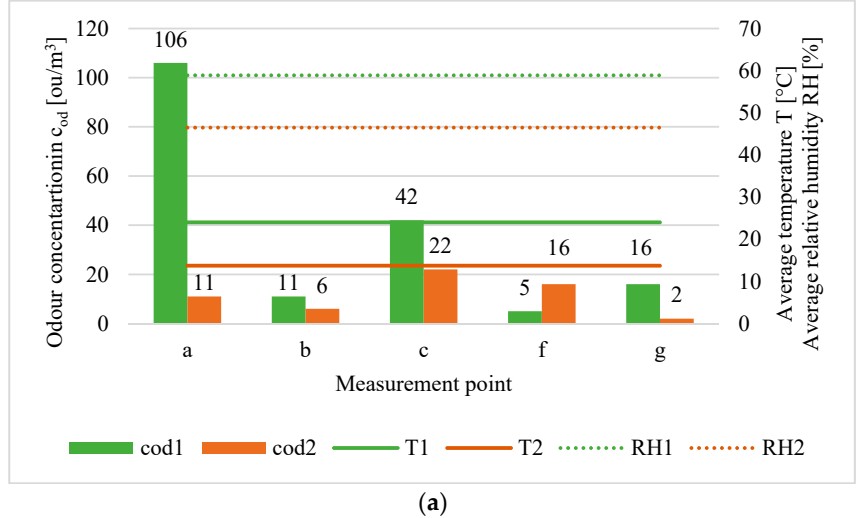

**Figure 6.** *Cont.*

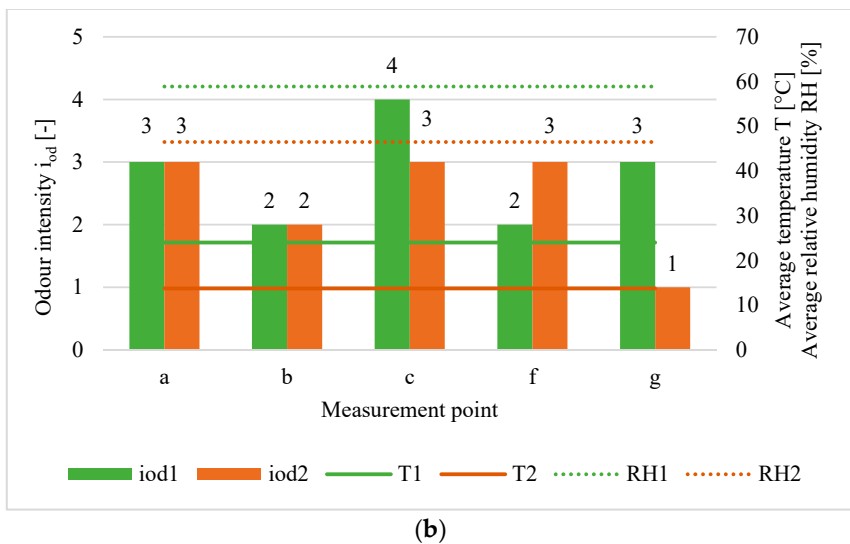

(**b**)

**Figure 6.** Dependencies between average air temperature and relative humidity and $c_{od}$ and $i_{od}$ in two measurement series at Wólka Rokicka biogas plant (28 August 2018 and 19 February 2019), (**a**) T, RH and $c_{od}$; (**b**) T, RH and $i_{od}$.

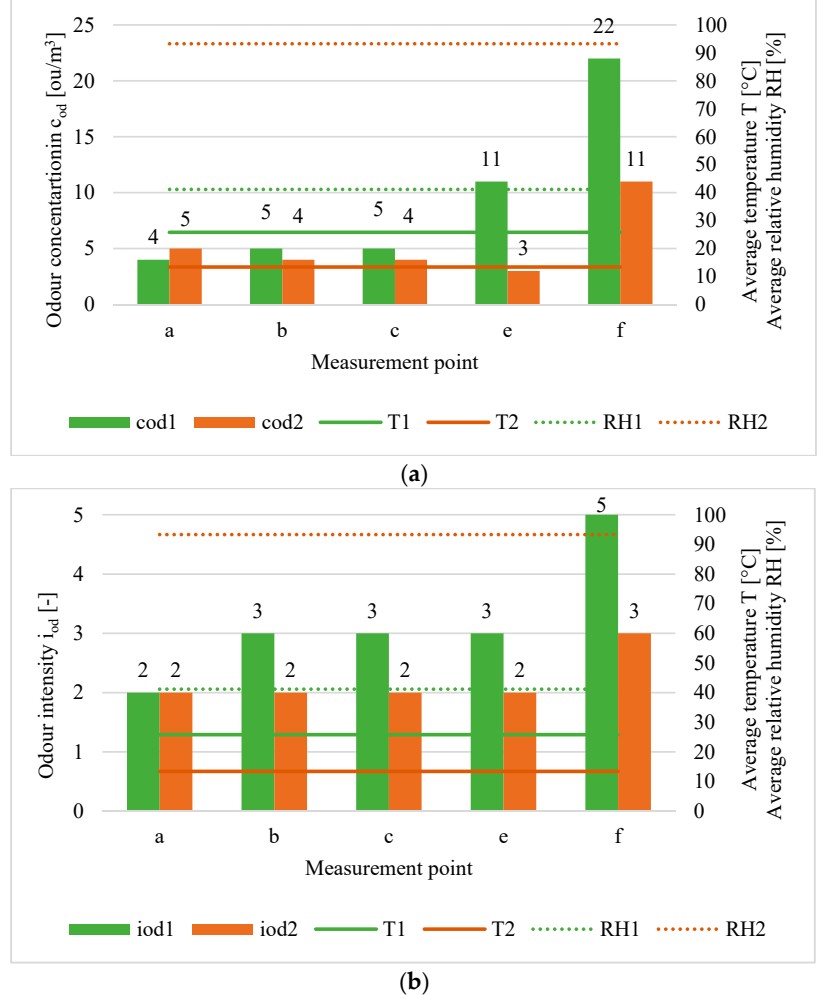

(**a**)

(**b**)

**Figure 7.** Dependencies between average air temperature and relative humidity and $c_{od}$ and $i_{od}$ in two measurement series at Biała Podlaska biogas plant (26 April 2019 and 14 May 2019), (**a**) T, RH and $c_{od}$; (**b**) T, RH and $i_{od}$.

At individual measurement points, $c_{od}$ varies from 2 ou/m$^3$ to 448 ou/m$^3$ depending on the kind of technological procedures. The highest values of $c_{od}$ is observed during digestate dewatering at A biogas plant (448 ou/m$^3$) and waste storage at the biogas plant at E biogas plant (106 ou/m$^3$), while the lowest ones are identified from the surface of the open biological filter in Wólka Rokicka (E) and Tychy (B) (respectively 2 ou/m$^3$ and 3 ou/m$^3$) and in the oxygen stabilisation of mixed waste sieve fraction (15–80 mm) plant in Biała Podlaska (F) (3 ou/m$^3$). It should be also noted that, at many odour sources, the measured values of $c_{od}$ is between 4 and 8 ou/m$^3$.

Analysing the differences between the results obtained at the same odour source, but in two different measurement series, it should be stated that in most cases, a higher $c_{od}$ is accompanied by a higher $i_{od}$. It is visible at all analysed biogas plants (A) at odour sources a, c, f, g; (B) at odour sources c,d; (C) at odour sources a, c, d; (D) at odour sources a, e, f, (E) at odour sources: c, f, g; and (F) at odour sources b, c, e, f (Figures 2–7). It should be noted that the results of $i_{od}$ are less varied between individual measurement series than $c_{od}$ results what is related to the methodology of determining the odour intensity (using the scale from 0 to 5, in steps of 1; Table 2).

### 3.3. Technological Factors

The impact of technological factors in the form of technological and operating processes and waste properties is observed on the basis of changes in VOC and NH$_3$ concentrations. Table 6 presents the differences between individual measurement series in the odorant concentrations at designated measurement points constituting the odour sources.

**Table 6.** Measurements of VOC and NH$_3$ concentration obtained in the pilot research at six biogas plants processing municipal waste.

| Biogas Plant | Parameter | No. of the Research Series | Odour Sources Identified at the Examined Biogas Plants | | | | | | |
|---|---|---|---|---|---|---|---|---|---|
| | | | a | b | c | d | e | f | g |
| A | VOC [ppm] | 1 | 0.20 ± 0.01 | 0.02 ± 0 | 0.20 ± 0.01 | 1.30 ± 0.01 | — | 0.21 ± 0.01 | 0.65 ± 0.01 |
| | | 2 | 0.81 ± 0.01 | 0.24 ± 0.01 | 0.53 ± 0.01 | 0.82 ± 0.01 | — | 0.60 ± 0.01 | 0.31 ± 0.01 |
| | NH$_3$ [ppm] | 1 | <1 ± 0 | <1 ± 0 | <1 ± 0 | 22 ± 1 | — | 1 ± 1 | 13 ± 1 |
| | | 2 | 1 ± 0 | <1 ± 0 | 1 ± 1 | 32 ± 2 | — | 4 ± 1 | 2 ± 1 |
| B | VOC [ppm] | 1 | 7.74 ± 0.02 | 6.04 ± 0.02 | 1.94 ± 0.01 | 1.83 ± 0.01 | — | — | 0.02 ± 0 |
| | | 2 | 3.81 ± 0.01 | 1.84 ± 0.01 | 1.37 ± 0.01 | 1.20 ± 0.01 | — | — | 0.13 ± 0 |
| | NH$_3$ [ppm] | 1 | <1 ± | <1 ± 0 | 5 ± 1 | 20 ± 1 | — | — | <1 ± 0 |
| | | 2 | 1 ± 1 | <1 ± 0 | 1 ± 1 | 9 ± 1 | — | — | <1 ± 0 |
| C | VOC [ppm] | 1 | 8.27 ± 0.02 | 4.34 ± 0.01 | 5.71 ± 0.02 | 6.41 ± 0.02 | nm | 0.14 ± 0 | nm |
| | | 2 | 4.26 ± 0.01 | 0.54 ± 0.01 | 1.66 ± 0.01 | 2.65 ± 0.01 | nm | 0.12 ± 0 | nm |
| | NH$_3$ [ppm] | 1 | 1 ± 1 | 1 ± 0 | 1 ± 1 | 2 ± 1 | nm | 1 ± 1 | nm |
| | | 2 | 2 ± 1 | 1 ± 0 | 1 ± 1 | 1 ± 1 | nm | 1 ± 1 | nm |
| D | VOC [ppm] | 1 | 0.89 ± 0.01 | 0.74 ± 0.01 | 2.38 ± 0.01 | 2.13 ± 0.01 | 1.33 ± 0.01 | 0.26 ± 0.01 | 1.55 ± 0.01 |
| | | 2 | 1.42 ± 0.01 | 0.60 ± 0.01 | 2.35 ± 0.01 | 0.20 ± 0.01 | 0.50 ± 0.01 | 0.58 ± 0.01 | 0.86 ± 0.01 |
| | NH$_3$ [ppm] | 1 | <1 ± 0 | <1 ± 0 | 1 ± 1 | 8 ± 1 | 3 ± 1 | 1 ± 1 | 14 ± 2 |
| | | 2 | <1 ± 0 | <1 ± 0 | <1 ± 0 | <1 ± 0 | 1 ± 1 | 1 ± 1 | 5 ± 1 |
| E | VOC [ppm] | 1 | 0.98 ± 0.01 | 2.50 ± 0.01 | 1.30 ± 0.01 | — | — | 0.96 ± 0.01 | 0.29 ± 0.01 |
| | | 2 | 0.72 ± 0.01 | 1.05 ± 0.01 | 1.40 ± 0.01 | — | — | 3.29 ± 0.01 | 0.26 ± 0.01 |
| | NH$_3$ [ppm] | 1 | 1 ± 1 | 3 ± 1 | 9 ± 1 | — | — | 1 ± 1 | 1 ± 1 |
| | | 2 | <1 ± 0 | <1 ± 0 | 1 ± 1 | — | — | 4 ± 1 | <1 ± 0 |
| F | VOC [ppm] | 1 | 0.14 ± 0.01 | 0.44 ± 0.01 | 0.50 ± 0.01 | — | 0.45 ± 0.01 | 3.32 ± 0.01 | nm |
| | | 2 | 1.40 ± 0.01 | 0.64 ± 0.01 | 0.63 ± 0.01 | — | 0.18 ± 0.01 | 0.47 ± 0.01 | nm |
| | NH$_3$ [ppm] | 1 | <1 ± 0 | 2 ± 1 | 4 ± 1 | — | 4 ± 1 | 20 ± 2 | nm |
| | | 2 | 1 ± 1 | 1 ± 0 | 1 ± 1 | — | 1 ± 1 | 9 ± 1 | nm |

Notes: The values of VOC and NH$_3$ concentrations which changes between individual measurement series coincided with the direction of $c_{od}$ changes are marked in grey. "—" sources do not occur at the premises of a given plant, due to the applied waste treatment technology, "nm" not measured.

The variation in VOC and NH$_3$ concentration between individual measurement series is primarily associated with the diverse technological properties of waste delivered to the plant and with different



compliance with the technological regime. The impact of the technological regime and the applied solutions has been discussed by Wiśniewska et al. [23].

### 3.4. Impact of Air Temperature and Humidity and Technological Factors on Odour Concentration and Intensity: Analyses

At four of the examined biogas plants (A, B, E, F) for two research series, bigger $c_{od}$ (at designated points) is accompanied by higher air temperatures (irrespective of differences between the temperatures in a range from 3 °C to more than 18 °C). The same relationships are also reported in other papers [34,36]. In the case of B biogas plant, despite small differences between mean temperatures of individual measurement series (at a level less than 3 °C), such dependencies are observed at almost all measuring points. In turn, at the biogas plant in Biała Podlaska in spite of a slight variation in $c_{od}$ values (from 3 ou/m$^3$ to 11 ou/m$^3$), the dependencies between $c_{od}$ at measurement sites and average air temperature are evidently the same. It is the air temperature that seems to have a major impact among environmental drivers on odour emission. This is particularly evident in relation to $c_{od}$ at A biogas plant. During the second measurement series, the average air temperature is lower by about 61% compared to the air temperature in the first series, and in individual points, lower $c_{od}$ values are also noted—lower by 33% (in b and g points) to 84% (in point a) as is graphically illustrated in Figure 2. This trend is not observed only at one point: point f (oxygen stabilisation). At other biogas plants, the tendency is similar, and the lower temperature is accompanied by lower $c_{od}$ values (compared in the same odour sources)—lower by 17% to 90%. This is illustrated graphically in Figures 3–7. The dependencies between $c_{od}$ and average air temperature are not so evident in Stalowa Wola. The variability of $c_{od}$ between the series of tests is probably determined more by the technological factors.

This tendency is less pronounced with respect to $i_{od}$ due to the aforementioned methodology that was used. At two examined biogas plants (B and F), dependencies which mirrored the observed $c_{od}$. A temperature relationship is also observed for $i_{od}$ and air temperature; a higher $i_{od}$ is found at higher air temperatures. At other biogas plants (A, D and E), $i_{od}$ is less dependent on atmospheric and ambient conditions because it was characterised by lower variability between the measurement series.

At A biogas plant, the difference between average air temperatures during the test series is too small (0.3 °C) to assess its effect on odour nuisance. It makes the comparison impossible.

At two biogas plants (A and F), a similar impact is observed of mean air humidity on $c_{od}$ and $i_{od}$ compared to the effect presented in the literature [35,36]—a higher $c_{od}$ and $i_{od}$ at lower air humidity. However, this relationship is observed in the case of differences in air humidity at a minimum level of about 20%. It should be noted that differences in average relative air humidity at the level of 10% to 14% (as measured respectively at C and E biogas plants) do not have a significant effect on $c_{od}$ and $i_{od}$. At two biogas plants, the differences between average air humidity during separate research series are too small to assess their effect on odour nuisance. The air temperature should be indicated as the main environmental factor visible primarily in relation to $c_{od}$. The strength of the correlation relationship between the odour concentration and air temperature was determined by means of Spearman's rank correlation coefficient. The choice of the applied coefficient results from the type of features, number of observations and shape of the relation. The correlation coefficient $r_{ho}$ for all measurements is $r_{ho} = 0.306$ and the significance level $\alpha = 0.022$. The results obtained for the source d (digestate dewatering plant) were not used for calculations. The implementation of dewatering process varies significantly between plants and is largely influenced by the technological solutions used there.

However, the influence of wind cannot be ignored. The greatest wind influence was observed in the case of F biogas plant for e source (oxygen stabilisation in the open air): in the first series $v_1 = 1.5$ m/s, $c_{od} = 22$ ou/m$^3$; in the second series $v_2 = 2.3$ m/s, $c_{od} = 11$ ou/m$^3$. The obtained relation (an increase in wind speed results in a decrease in odour concentration) is consistent with the results of tests carried out by Sakawi et al. [44]. This impact should be taken into account during further testing.

The obtained results of air temperature and humidity impact on odour emissions are in accordance with results presented by other authors; but it should be emphasised that the scope of the research

reported in the literature has not allowed for unequivocal, forecastable and obvious assessment of the discussed dependencies. Thus, this paper has contributed significantly to the subject. Knowledge of the impact of various factors (meteorological, ambient and technical) on the odour emissions and nuisance of waste treatment plants will permit to apply corrective actions to minimise them.

The conducted research also shows the impact of technological processes, operating regimentation and changes in waste properties. The variability of these factors can be observed on the basis of the results of VOC and $NH_3$ concentrations, changing between separate series. In most cases, the increase in VOC or $NH_3$ concentration is accompanied by an increase in $c_{od}$ (in five out of six installations—B, C, D, E and F—at all measuring points (E and F biogas plants)) or in most of them (Table 5). It demonstrates the effect of technological factors on the odour impact of biogas facilities. The influence of technological factors takes place in both the mechanical and biological parts of the installations. Especially at B and C biogas plants (in Tychy and Promnik), the impact of technological factors on the odour effect is significant (higher $c_{od}$ at higher VOC or $NH_3$ concentration), because in these cases the research was accompanied by small differences in air temperature and humidity between individual measurement series, so their influence needs to be omitted.

The increase in VOC or $NH_3$ concentration is not always accompanied by an increase in $c_{od}$. In these cases, a dominant influence of atmospheric factors is observed (according to the previously presented relationships). It is particularly evident at A biogas plant (in Jarocin), especially at the measuring points in the mechanical part of the installation. In turn, the biological part of this plant (point f: oxygen stabilisation) shows the prevailing impact of technological factors over atmospheric agents in relation to odour emissions.

The highest VOC concentrations are recorded in the plants of waste storage (7740 ppb), pre-treatment (6040 ppb and 5710 ppb) and digestate dewatering (6410 ppb) at B and C biogas plant (points a, b, c, d according to Table 6). Among the places where the highest $NH_3$ concentrations are observed for digestate dewatering plant (32 ppm, 22 ppm and 20 ppm), installation of oxygen stabilisation (20 ppm) and open biofilter surface (13 ppm and 14 ppm) should be mentioned (points d, f, g at A, B, D, F biogas plants according to Table 6). Further research should indicate the dominance of some element in the technological line in this respect. Analysing the presented results, it should be stated that at different measuring points (in the technological line) different odorants (VOC or $NH_3$) have the main impact on the level and changes of odour concentration. At points a, b, d, f (waste storage, pre-treatment, digestate dewatering and oxygen stabilisation), more VOC concentration changes accompanying $c_{od}$ changes are observed and marked in grey in Table 6. In other elements of the installations, the influence of $NH_3$ or both odorants dominate. Further research should indicate which odorant is responsible for higher odour emission.

## 4. Conclusions

In this study, the results of pilot research conducted at six biogas plants processing municipal waste in Poland are presented. They show the relations between odour intensity and concentration and the occurring meteorological and ambient conditions and technological factors. The novelty and scientific contribution presented in this paper are related to the analyses of technological aspects and air temperature and humidity impact on odour emissions in municipal waste biogas plants. The literature review shows, that so far, such analyses were conducted mainly in the field of sewage treatment, swim housing, landfilling and animal husbandry. The impact of meteorological and ambient conditions is identified by measuring air temperature and relative humidity and observing their changes. The impact of technological factors was identified by measuring odorant concentration (volatile organic compounds and ammonia) and observing their changes between individual measurement series. The main conclusions and contributions of this work can be summarised as follows:

1.  At most analysed biogas plants, significant relations are observed between odour concentration and air temperature: higher air temperature is accompanied by bigger odour concentrations.

These relationships are noted even at small differences in both air temperature and odour concentration.

2. The relationships between relative air humidity and odour concentration (bigger odour concentrations at lower air humidity) are observed in the case of air humidity differences above 20%. To capture the full impact of air humidity, research should be conducted under more diverse atmospheric conditions.

3. In planning processes for municipal waste biogas facilities, great care in the choice of technological solutions, technical and organisational measures is called for to reduce the impact of unfavourable atmospheric conditions on odour emissions.

4. At most analysed biogas plants, the influence of technological factors on odour emissions takes place and is clearly noted. It is particularly distinct in the biological part of the installations, but it also takes place in the mechanical part (during waste pre-treatment).

5. The highest VOC concentrations are mainly associated with unit operations related to mechanical waste processing, such as storage, pre-treatment and digestate dewatering. High ammonia emissions are also associated with the elements of the mechanical part of the installations (waste preparation and digestate dewatering), but in addition, they may be accompanied by digestate aerobic stabilisation and the operation of deodorisation facility. Further research should indicate the dominance of some element in the technological line in this respect and show which odorant is responsible for higher odour emission.

6. The experimental results provide an important contribution to the search for more efficient methods for reducing the odour nuisance of biogas plants, which is important for residents of the surrounding areas.

7. Recommendations for further research include subsequent investigating emission of odorants characteristic for waste management, such as volatile organic compounds, ammonia or hydrogen sulphide. Further works should focus on clarifying when meteorological and ambient conditions have a greater impact on odour concentration and intensity, and when technological factors are more important and which of them have a dominant influence.

**Author Contributions:** M.W.: data curation, investigation, writing—original draft preparation; A.K.: conceptualization, methodology, supervision; K.L.-S.: writing—review and editing, visualisation, validation. All authors have read and agreed to the published version of the manuscript.

**Funding:** Work was co-financed by the Dean's Grant (No. 504/03693/1110/42.000100): "Identification and characteristics of the sources at biogas plants processing municipal waste".

**Conflicts of Interest:** The authors declare no conflicts of interest.

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
