# Peer review of "Odour Emissions of Municipal Waste Biogas Plants—Impact of Technological Factors, Air Temperature and Humidity"

_applsci, doi:10.3390/app10031093_

Round 1

Reviewer 1 Report

Interesting research on the evaluation of emissions from biogas plants according to different aspects; it should rather be revised and detailed to make it more comprehensive. The introduction is rather dispersive by mixing experience from very different plants and should be reorganized; the initial part on the description of the reliefs relating to the characterization of the plants, of the pre-tritment and effluent management systems should be deepened, perhaps coding them in order to avoid mentioning them only by place in the rest of the text and therefore obliging to always go backwards to see the type of related plant. Some results can be considered rather obvious; some of the considerations on future studies should be addressed in this.

Reviewer 2 Report

The study proposed by Wiśniewska et al. aimed at understanding the sources and drivers of odour emissions at 6 municipal waste biogas plants in Poland. The authors showed an influence of temperature, humidity and technological factors. The paper contains valuable results that could be considered for publication, but the results are poorly presented and the discussion is not enough consistent. The author statements need to be more evidenced, and the discussion required really extended before considering the paper for publication in applied sciences. In addition, the experiment seems to be conducted in appropriate conditions, but more details/precisions are needed to confirm this point and better understand the results. My comments are listed below to try to help the authors improve their article.

Comments:

C1. Both past and present are used in the paper (see for example L.10-11 : “the paper has presented…. It shows … “, or L.111-1122) keep consistency through the paper.

In addition, a sentence introducing the background is needed at the beginning of the abstract.

I also think that the abstract is longer than the recommendation of the journal (no more than 200 words)

C2. L.93-96 : not clear, please rephrase

C3. L. 114 : “the droplets of water vapour” : no sense as droplets refers to liquid phase, and vapour refers to gas phase. Please rephrase.

C4. L.118-121 : Same sentence as the previous one, please chose only one of them.

C5. L.129-139 : In my opinion, this part is not necessary and redundant with the above introduction where the authors already pointed out why the present study is important. Some of the sentences could be moved in the introduction above.

C6. L.141-144 : Again, this was more or less already stated between L. 117 and 128. I recommend to remove this.

C7. L.149 : It is impossible to control wind speed and direction. I guess the authors wanted to say that they checked/measured wind speed and direction… If true why there is no value of wind speed and direction later in the paper (especially considering that wind is an important parameter as suggested in the introduction)?

Please give the exact model of the Weather Meter

C8. L.167 : Please expressed what does Ztak and Znie stand for

C9. L.173 : Please give more details about the sensor (functioning, type of sensor used, time resolution, limit of detection, etc. ). In addition, it would be interesting to have a short discussion about potential artefacts and the response of the sensor to the different class of VOCs.

C10. L146 : Give more details about the two measurement series : Have they been performed at the time of the day ? During similar conditions (rainy ? windy ?) ?

C11. Section 2.2 : The analysed biogas plants are not enough detailed, the reader have no ideas about their differences and if they are comparable. In addition, there no information about the measurement period (day or night ? season? time of day) This section should be developed.

C12. Table 3: why source d/serie 2 is the only measurements given with uncertainties ? What do these uncertainties correspond (standard deviation ?) ? Please add uncertainties for all or none measurements.

C13. L203-205: What is the reason for such different temperatures between series 1 and 2 ? Same remark for relative humidity (L.208-211). As this section is included in Results and discussions, the authors have to not only present the results but also explain/discuss them

C14. Figure 2 to 7 : The text never referred to these figures because there is no real discussion about the results presented in these figures... In addition, some numbers overlapped with lines, making them difficult to read…. Please upgrade figure design, add a real discussion about the results presented in these figures, and referred to them when it is appropriated. Otherwise, it means that figures 2 to 7 are useless.

C15. L251-252 : Please discuss this statement based on the Figures

C16. Table 4 : What does the +- correspond ? In addition, what is the significance of a +-0  (see for example source f or biogas plant B)?

C17. L260-261 : I cannot see anything marked in grey….

C18. L.270 -279 : The authors explained that cod is related to temperature at 4 biogas plants. This is not really supported by the current Figures in the paper. A plot of cod as a function of temperature would help to support this hypothesis and evidenced the author statements. Moreover, statistics such as r2 can be investigated to provide more reliable correlations/results.

C19. L.301-319 : This discussion is interesting but is very limited and need to be extended. Below is a non-exhaustive list of questions that need to be more extensively explored:

What is the main environmental driver of Cod/Iod ? Is it T or RH ? Bring more evidence about the relation between environmental parameters and cod/iod. Also include wind speed and direction if you measured them (see my comment above)

Ammonia or VOC : which one is the main contributor to the odour measurement ?  

Is there a source (maybe digestate dewatering plant ?)  emitting more VOCs than others ? This is not enough discussed and not really stated in the manuscript. The author only state that technological factors affect odour emissions, but do not precise which one is responsible for higher/lower emissions. This point should help to make future recommendations

C20. Conclusion: Too long and repetitive. Need to be re-written with only the main findings, after considering the above comments.

Editing comments

L.66 : for instance

L.71 : replace “The paper” by “A previous paper”

L.75 : state that VOC stand for Volatile Organic Compounds

L.80 : Please add a reference related to your statement

L.113 : “results”

L.115 : “conducted a similar research and obtained similar results”

L.146 : replace “the research” by “the study”

Table 3 : change t to T, as t usually referred to time (applied all along the manuscript). Same for h, it usually referred to hours. Change for RH (Relative Humidity)

L.197-198 : Not clear. Please rephrase

L.242 and through : cod ->

Round 2

Reviewer 1 Report

The paper has improved, some aspects may be more developed but overall publishable

Reviewer 2 Report

I think that the revised version of the paper can now be considered for publication.